# Effects of Lactic Acid Bacteria Reducing the Content of Harmful Fungi and Mycotoxins on the Quality of Mixed Fermented Feed

**DOI:** 10.3390/toxins15030226

**Published:** 2023-03-16

**Authors:** Yifei Guan, Haoxin Lv, Guofang Wu, Jun Chen, Min Wang, Miao Zhang, Huili Pang, Yaoke Duan, Lei Wang, Zhongfang Tan

**Affiliations:** 1School of Agricultural Sciences, Zhengzhou University, Zhengzhou 450001, China; guanyifei0904@163.com (Y.G.); chenjun6377@163.com (J.C.); wangmin509867@163.com (M.W.); miaozhang@zzu.edu.cn (M.Z.); pang@zzu.edu.cn (H.P.); duyk@nwafu.edu.cn (Y.D.); 2School of Food and Strategic Reserves, Henan University of Technology, Zhengzhou 450001, China; lvhaoxin0129@126.com; 3Plateau Livestock Genetic Resources Protection and Innovative Utilization Key Laboratory of Qinghai Province, Key Laboratory of Animal Genetics and Breeding on Tibetan Plateau, Ministry of Agriculture and Rural Affairs, Qinghai Academy of Animal and Veterinary Medicine, Qinghai University, Xining 810016, China; jim963252@163.com

**Keywords:** lactic acid bacteria, antifungal activity, fermented feed, mycotoxin

## Abstract

The contamination of fermented feeds and foods with fungi and mycotoxins is a major food safety issue worldwide. Certain lactic acid bacteria (LAB), generally recognized as safe (GRAS) fermentation probiotics, are able to reduce microbial and mycotoxins contamination. In this study, *Lactiplantibacillus* (*L.*) *plantarum* Q1-2 and *L. salivarius* Q27-2 with antifungal properties were screened as inoculants for mixed fermenting feed, and the fermentation and nutritional qualities, microbial community, and mycotoxins of mixed fermented feed were analyzed at different fermentation periods (1, 3, 7, 15, and 30 days, respectively). The findings indicated that the utilization of Q1-2 and Q27-2 strains in fermenting feed led to a decrease in pH and an increase in lactic acid concentration and the proportion of *Lactiplantibacillus*, while effectively restraining the proliferation of undesirable microorganisms. In particular, Q1-2 reduced the relative abundance of fungi including *Fusarium* and *Aspergillus*. Compared to the control group, the Q1-2 and Q27-2 groups reduced aflatoxin B1 by 34.17% and 16.57%, and deoxynivalenol by up to 90.61% and 51.03%. In short, these two LAB inoculants could reduce the contents of aflatoxin B1 and deoxynivalenol to the limited content levels stipulated by the Chinese National Standard GB 13078-2017. These findings suggest that the LAB strains of Q1-2 and Q27-2 have potential applications in the feed industry for the mitigation of mycotoxin pollution, thereby enhancing the quality of animal feed.

## 1. Introduction

Microbial contamination includes bacterial, viral, and fungal contamination, as well as contamination by their toxins. According to a report by the Food and Agriculture Organization of the United Nations (FAO), hundreds of billions of dollars are lost across the world each year due to fungal and toxin infestation of crops, resulting in loss of food value and consequent significant economic losses [1]. Fungi are generally found in soil, air, and plants and have the ability to contaminate a variety of foods and animal feeds [2]. Mycotoxins are poisonous secondary metabolites produced by filamentous fungi mainly belonging to the *Fusarium*, *Aspergillus,* and *Penicillium* genera, which contaminate various food crops. Mycotoxins have the ability to enter the body through contaminated animal feed and food, posing a significant threat to human and animal health. Upon ingestion, these toxic secondary metabolites produced by fungi can cause detrimental effects on vital organs such as the liver, kidney, and nervous system [3,4]. As secondary products of these fungi, mycotoxins could cause direct adverse effects on animal and human health, with aflatoxin B1 (AFB1) and deoxynivalenol (DON) considered to be the most influential. AFB1 is widely recognized as the most toxic and carcinogenic mycotoxin, which causes cancer through the cross-linking of DNA and guanine. Compared to other mycotoxins, AFB1 exhibits a high rate of absorption through the gastrointestinal tract, with up to 80% of ingested AFB1 being absorbed. Acute poisoning caused by the excessive consumption of AFB1 can result in symptoms such as acute hepatitis, hemorrhagic necrosis, hepatocellular steatosis, and bile duct hyperplasia. Furthermore, chronic toxicity can arise even from exposure to small amounts of AFB1, resulting in growth disorders, fibrous lesions, and the hyperplasia of fibrous tissue [5]. DON is one of the most serious and widespread mycotoxins contaminating feed in China [6]. When feed is contaminated, its color, flavor, and nutrients change, and its nutritional value is substantially reduced; different species and different concentrations of mycotoxins can cause various adverse phenomena such as growth retardation and reduced immunity in animals [7]. Mycotoxin surveys conducted in Poland have shown that up to 95% of feeds contain at least one mycotoxin [8]. AFB1 was detected in up to 87.8% of feeds and their raw materials in China, while DON was detected in over 95% of feeds [9]. Mycotoxins are not only found in feed and food; this hazardous chemical is also frequently found in raw materials [10]. These data could indicate that the infestation of feed and its raw materials with mycotoxins is quite serious. Adding natural probiotics to feeds can not only control the growth of harmful microorganisms and mitigate mycotoxins contamination but also maintain the original composition and taste of feeds. The addition of biological agents is environmentally friendly, regulates the balance of intestinal flora, and strengthens the immunity of the organism, which is incomparable to physical sterilization and chemical additives [11].

In recent years, lactic acid bacteria (LAB), recognized as a safe and qualified additive by the Food and Drug Administration (FDA), have been shown to inhibit fungal growth and degrade mycotoxins [12]. LAB could inhibit fungal spore germination and mycelial growth by competing for growth space [13] and by secreting nutrient-rich microbial active substances [14]. Generally, nutrient-rich substrates readily support microbial growth, but *L. rhamnosus GG* and *L. paracasei* could limit the growth of yeasts and fungi [15]. Antifungal LAB produce active metabolites such as lactic acid, acetic acid, cyclic dipeptides, phenylacetic acid, hydroxy fatty acids, and 3-hydroxy propionaldehyde, which have been shown to be associated with the antifungal effect of LAB [16]. Parappilly et al. [17] found that LAB produced organic acids, diacetyl, and other antibacterial substances to inhibit *Aspergillus flavus*. Crowley et al. [18] prevented the reactivation of fungi during rearing by adding LAB ferments with antifungal activity during the rearing process. LAB degrade mycotoxins mainly by enzymatic action or by the adsorption of microbial cell walls. *L. bulgaricus*, *L. paracasei*, and *L. plantarum* could be used as additives in fermented feeds because of the high protein hydrolysis activity of these three strains [19]. The mechanism of mycotoxin adsorption by LAB is ion exchange through the cell wall. The binding of substances in the cell wall to mycotoxins via hydrogen bonding, ionic bonding, and hydrophobic interactions provides potential binding sites [20]. Hernandez-Mendoza et al. [21] found that the complex of LAB with AFB1 was stable and that the complex formed by *L. casei* was the most stable. LAB were also able to bind AFB1 in the gut, thus reducing the toxicity of AFB1.

As a type of forage with high protein content, alfalfa is widely planted all over the world because of its good palatability and strong stress resistance. The fermentation of alfalfa could preserve nutrients and prolong the shelf life of feed. However, the low water content and small LAB attachment amount makes it difficult to directly ferment natural feed. Adding LAB to fermented feed could improve the palatability and fermentation quality of feed [22]. Jiang et al. [23] reported that *Pediococcus pentosaceus* could enhance the quality of feed fermentation by lowering the pH value and increasing the lactic acid content. Muck et al. [24] demonstrated that LAB could rapidly increase the range of organic acids, especially lactic acid, and improve the aerobic stability of the feed. The effect of adding LAB to alfalfa fermented rations on bacteria has been extensively studied. Nevertheless, fermentation characteristics and microbial community changes in mixed fermented feed, especially changes in fungi and mycotoxins, need to be studied further. Hence, the main objective of this study was to obtain LAB with fungal inhibition and the reduction of mycotoxins and to apply them to mixed fermented feeds. The aim is to provide an excellent primary raw material for biological preservatives used in feed as well as food production.

## 2. Results

### 2.1. Screening of LAB with Antibacterial, Antifungal Activity, and Mycotoxin Adsorption Ability

Ninety-seven strains of LAB were isolated from the intestinal feces of the Bamei pig. Four strains with antibacterial activity against *Escherichia coli* were screened, and further broad-spectrum antibacterial tests were carried out. As can be seen from Figure 1 and Figure 2, and Table 1, strains Q1-2 and Q27-2 had excellent antibacterial effects on both bacteria and fungi. The adsorption rate of mycotoxins was significantly higher than that of the other strains. As shown in Table 2, strains Q1-2 and Q27-2 were ultimately selected for fermentation feed. In addition, both the Q1-2 and Q27-2 strains showed an increase in mycotoxins adsorption after heat inactivation. Of the two, heat-inactivated Q1-2 showed a significant increase in the adsorption rate of AFB1. The adsorption rate of DON by heat-inactivated Q27-2 was also significantly increased.

### 2.2. 16S rRNA Gene Sequence Analysis

According to the results of antibacterial activity, a 16S rRNA gene analysis was performed on Q1-2 and Q27-2. BLAST was applied in order to compare the 16S rRNA gene sequences of the superior strains Q1-2 and Q27-2 in GenBank. The results showed that the 16s rRNA gene sequences of the two strains and multiple strains of *Lactiplantibacillus* were identified by more than 99%, and the phylogenetic tree was constructed by the neighbor-joining method (Figure 3). The two strains were placed in a cluster made up of the genus *Lactiplantibacillus*, so the strain Q1-2 was identified as *L. plantarum* and the strain Q27-2 was identified as *L. salivarius*.

### 2.3. Growth Curve of Selected LAB Isolates

The growth curves of Q1-2 and Q27-2 cultured at 37 °C are shown in Figure 4. The growth curves of the two strains showed sluggish, logarithmic, and stable stages. The period of 0–4 h was the slow stage, 6–12 h was the logarithmic growth stage, and at 12 h the stable bacterial growth stage was entered.

### 2.4. Physiological and Biochemical Characteristics

As shown in Table 3, Q1-2 and Q27-2 have strong salt tolerance, low-temperature resistance, and acid tolerance. Q1-2 has strong alkaline tolerance. The two screened strains grew weakly in a high-temperature environment and did not have high-temperature tolerance.

### 2.5. Fermentation Quality and Chemical Composition

The pH value changes during fermentation are shown in Figure 5. All groups had lower pH values than the initial feed samples at 30 d of fermentation. Compared to the control group, the pH value of the treatment group supplemented with Q1-2, and Q27-2 rapidly decreased 3 days before fermentation and recovered 30 days after fermentation. In the control group, the pH value showed a slowly decreasing trend. On all fermentation days, the pH values of all treatment groups were significantly lower compared to those of the control group.

The fermentation quality and chemical composition of feedstuffs were measured at five time points: 1, 3, 7, 15, and 30 days after the onset of fermentation. The fermentation indexes of the different treatment groups are shown in Table 4. The amounts of lactic acid, acetic acid, and ammonia nitrogen in the treatment group (T) were significantly affected (*p* < 0.05). The lactic acid content of the three treatment groups showed an overall increasing trend. For 30 days, the content of lactic acid in the control group was significantly lower than that in the treatment group, while the range of acetic acid and ammonia nitrogen was substantially higher than that in the other treatment groups. At the same time, only propionic acid and butyric acid were detected in the control group after 30 days of fermentation.

The chemical composition of the fermentation feed was sampled at 1, 3, 7, 15, and 30 days. The chemical composition of different treatment groups is shown in Table 5. Significant effects were observed for both the treatment group (T) and fermentation days (D) as well as the interaction between the treatment group and fermentation days (T × D) on the CP content (*p* < 0.05). The CP content of the control group was consistently higher than in the other treatment groups across all time points. The contents of NDF and ADF showed a decreasing trend over time.

### 2.6. Microbial Diversity and Community Analysis during Fermentation

The principal co-ordinates analysis (PCoA) diagram explains the changes in microbial communities by analyzing the beta diversity index, as shown in Figure 6. Greater similarity in community composition among samples would result in a more clustered appearance in the principal component analysis diagram. In terms of bacterial composition, there was a certain distance between each treatment group; that is, there was a certain difference in bacterial composition between the control group and the treatment group. In terms of fungal composition, the treatment group with Q1-2 was significantly separated from the other groups, showing a significant difference.

The relative abundance of bacteria and fungi during feed fermentation is shown in Figure 7. The bacterial community dynamics of fermented feed at the phylum level are shown in Figure 7a. In the original sample, *Firmicutes* (22.24%) and *Cyanobacteria* (62.59%) were dominant at the gate level. In the fermentation process, *Firmicutes* and *Cyanobacteria* were dominant in all groups, but the specific community groups were affected by fermentation treatment. The community composition changed with the fermentation process. The Q1-2 group showed a significantly higher relative abundance of *Firmicutes* compared to other treatment groups on the 15th day, while the control group exhibited a significantly higher relative abundance of *Bacteroidota* compared to other treatment groups. At the end of fermentation, the relative abundance of *Firmicutes* in the Q1-2 group was lower than that in the different treatment groups. However, on the 30th day, the relative abundance of *Firmicutes* increased in all treatment groups compared to the unfermented sample.

Figure 7b illustrates the dynamics of bacterial community composition in fermented feed at the genus level. *Chloroplast* (62.5%) had the highest relative abundance in unfermented diets. After fermentation, the relative abundance of *Lactiplantibacillus* increased in all groups. After 7 days of fermentation, the *Lactiplantibacillus* in the Q1-2 and Q27-2 groups was higher than in the control group. On the 15th day of fermentation, the relative abundance of *Lactiplantibacillus* in the treatment group with the strain Q1-2 added was 61.15%, significantly higher than that in the other treatment groups, and the relative abundance of *Lactiplantibacillus* decreased with fermentation. *Parabacteroides*, a genus of propionic acid-producing bacteria with a high relative abundance (38.14%), appeared in the control group at 15 days, while *Prevotella* appeared in the treatment group with a relative abundance of 6.89% at 30 days after the addition of strain Q1-2.

The relative abundance at the phylum level of the fungal community during fermentation is shown in Figure 7c, and *Ascomycota* and *Basidiomycota* were dominant in different treatment groups. The Q1-2 group had a lower relative abundance of *Ascomycota* compared to the other treatment groups, except on the 30th day. Figure 7d displays the relative abundance of fungi at the genus level throughout the fermentation process. In the unfermented samples, *Fusarium* was dominant, with a relative abundance of 60.53%. The relative abundance of fungi in the control group was similar to that in the treatment group with strain Q27-2. The Q1-2 group had a significantly lower relative abundance of *Fusarium* compared to other treatment groups at all time points. At 15 d, *Saccharomyces* (4.27%) appeared in the treatment group with strain Q1-2, and its relative abundance was significantly higher than that in the other treatment groups. At day 30 of fermentation, the *Aspergillus* content of the Q1-2 group was lower than the rest at any time point.

### 2.7. The Contents of Mycotoxins Aflatoxin B1 and Deoxynivalenol during Fermentation

The changes in mycotoxins in the feed during fermentation are shown in Figure 8. The Q1-2 and Q27-2 groups showed lower levels of the mycotoxins AFB1 and DON compared to the control group. The contents of two mycotoxins in the Q1-2 group were lower than in the Q27-2 group, which was the lowest among the three treatment groups. As shown in Figure 8a, AFB1 content showed a downward trend in the first 15 d and continued to decline in the control group at 30 d, but increased in the treatment group with added strain. The change in DON content is shown in Figure 8b. The DON content in all treatment groups showed a decreasing trend with time, except for a slight increase in the Q1-2 group between 15 d and 30 d.

The relationship between fungal abundance and mycotoxins is shown in Figure 9. In the LAB-added treatment group, there was a positive correlation between the DON content and the relative abundance of *Fusarium*. The AFB1 content was positively correlated with *Aspergillus* and *Vishniacozyma* in all treatment groups, while *Saccharomycopsis* was negatively correlated with both AFB1 and DON content.

## 3. Discussion

There were a significant number of harmful microorganisms in the mixed fermented feed prepared from alfalfa, especially *Enterobacteriaceae* [25]. *E. coli*, a common cause of diarrhea, was also used as an indicator strain for primary screening. Starting with the *Enterobacteriaceae*, strains with an inhibitory effect on *E. coli* were screened from the intestines and feces of Bamei pigs, and their broad-spectrum antibacterial activity was then further analyzed. Mixed fermented feeds were often infected by fungal and mycotoxin contamination, and biological control—as one of the most effective treatments for fungi and mycotoxins—has been widely reported as a research hotspot in recent years [26]. Therefore, in this study, LAB with broad-spectrum antibacterial activity were tested for their antimicrobial spectrum against common fungi as well as for their mycotoxin adsorption ability. LAB had an inhibitory effect on molds, with *Lactiplantibacillus* spp. displaying the highest antifungal activity against molds of all the LAB isolated [27]. Gomaa et al. [28] showed the significant inhibition and degradation of molds and mycotoxins by treatment with *Lactiplantibacillus* spp., and the biomass of mycelium and mycotoxin production decreased significantly, with a degradation rate of 96.31%. In the present study, the toxin reduction rate of the two strains of LAB was only 48.2–63.4%, which may be due to the fact that the two strains of LAB reduced toxins differently. Both strains showed an increased efficiency in mycotoxin adsorption after heat inactivation, which is consistent with the study by El-Nezami et al. [29]. The ability of the LAB to adsorb mycotoxins after heat inactivation suggests that the binding of mycotoxins by LAB occurs on the cell wall. The strains Q1-2 and Q27-2 were selected as alternative strains for further research. In terms of strain identification, the physiological and biochemical indexes of strain Q1-2 and strain Q27-2 were tested in this study. Both LAB strains are acid tolerant and meet the requirements of the feed fermentation process [30]. In addition, the 16S rRNA PCR result was also used for identification. By comparing the amplification of the 16S rRNA gene through PCR, strains Q1-2 and Q27-2 were identified as *L. plantarum* and *L. salivarius*, respectively. Studies have shown that LAB are an essential promoter of lactic acid fermentation, and adding LAB to fermented feed could effectively increase the initial load of LAB [31]. Therefore, strains Q1-2 and Q27-2 were incorporated into the mixed fermented feed as feed additives for further study.

In the mixed fermented feed experiments, the decrease in pH during fermentation was used to indicate microbial activity and the smooth progress of fermentation [32]. In this experiment, the pH value was significantly reduced compared to that before fermentation, especially in the treatment group; with the addition of strain Q1-2 and strain Q27-2, the feed pH value was rapidly reduced to below 4.0 within three days. The rapid drop in pH value means that LAB proliferate and ferment to produce more lactic acid, which inhibits the growth of harmful bacteria. If lactic acid is insufficient, it will provide favorable opportunities for the growth of *Clostridium*, and the proliferation of *Clostridium* leads to the hydrolysis of sugars and proteins in the feed. Therefore, the high-quality fermentation feed depends on the speed of pH value’s decline [25], and obviously, this requirement is met by adding the strains Q1-2 and Q27-2. However, the pH value of the treatment group with the added strain showed a rise at 30d. Tian et al. [33] also reported similar results, possibly due to the decrease in soluble sugar content in the feed, which delayed the fermentation process.

Using LAB to supplement feed has been shown to be an effective approach for enhancing feed quality. The addition of the Q1-2 or Q27-2 strain had a significant positive effect on the fermentation index of fermentation feed. This study found a significant decrease in pH value in the treatment group compared to the control group, which was attributed to the higher levels of lactic acid in the treatment group. It has been shown that lactic acid is usually the main factor in pH value reduction in high-quality fermentation feed [34]. High lactic acid content also inhibited the growth of harmful bacteria or fungi, consistent with the excellent antibacterial activity of strains Q1-2 and Q27-2. The fermentation of *Clostridium* produces butyric acid which greatly decreases the palatability of the feed, although this is not expected to happen in high-quality feed. No butyric acid and propionic acid were detected in the treatment group supplemented with LAB in the late fermentation stage, which may be because the growth of *Clostridium* was inhibited by the LAB additive and *Clostridium* could not metabolize normally [25]. As the growth of *Clostridium Difficile* is inhibited, the decomposition rate of protein also slows down, thus reducing the content of ammonia nitrogen in fermentation feed, which is consistent with the findings of Silva et al. [30]. In addition, the Q1-2 and Q27-2 groups showed a significant decrease in ammonia nitrogen content without the detection of butyric acid, which verified this conclusion.

In this study, the number of days of fermentation had a significant effect on CP content (*p* < 0.05). The CP is an important indicator of mixed fermentation feeds, which is frequently hydrolyzed by protease and microbial activity into peptides, amino acids, and ammonia. As fermentation time increased, the CP content decreased, which could be attributed to the consumption of organic matter by beneficial bacteria through respiration, resulting in the release of CO_2_ and water and a reduction of the total amount of products, which is consistent with the findings of Du et al. [35]. Moore et al. [36] showed that the smaller the content of NDF and ADF, the better the quality of the feed, and the two were negatively correlated. The quality of mixed feeds was improved, as indicated by the decrease in NDF and ADF content during fermentation in the present study.

*Firmicutes* is one of the dominant phyla in the fermentation process, as most of the bacteria involved in lactic acid fermentation belong to *Firmicutes* [37]. The relative abundance of *Firmicutes* increased with the fermentation of feeds. However, in the Q1-2 group, the relative abundance of *Firmicutes* at 30 days was significantly lower than that at 15 days, suggesting that the rate of fermentation slowed down. This could be the reason for the increase in pH at the end of fermentation for the Q1-2 group. At the end of fermentation, the treatment group inoculated with *Lactiplantibacillus* had a significantly higher relative abundance of *Lactiplantibacillus* at the genus level compared to the control group, and *Lactiplantibacillus* was one of the dominant genera. *Lactiplantibacillus* is beneficial to lactic acid fermentation and is positively correlated with feed quality. *Parabacteroides* was present in the control group, and *Parabacteroides* is a propionic acid-producing organism [38], which may be one of the reasons why propionic acid was only detected in the control group. As shown in the PCoA diagram, some of the treatment groups separated, indicating that the addition of strains changed the composition of bacterial colonies and significantly affected the succession of the microbial community, which may be the reason that the pH value of the control group was significantly higher than the treatment group.

The PCoA diagram in Figure 6b revealed that the added strain Q1-2 was distinctly separated from the other treatment groups, indicating significant differences in fungal structure between the added strain and the other groups. This pattern was also reflected in the fungal species abundance histogram. Fungi and mycotoxins are an integral part of the study of fermented feeds. Studies have shown that harmful fungi could be present on the surface of crops or in stored feeds and grains as soon as the right temperature and moisture conditions are reached [39]. Feed contaminated with common mycotoxins such as AFB1 and DON, which are produced by harmful fungi, can cause severe damage to the immune system of livestock once it has been consumed, thus endangering the health of the animals [40]. *Ascomycota* was the absolute dominant strain at the level of phylum fungi, while the relative abundance of *Fusarium* was significantly lower in the Q1-2 group compared to the other treatment groups at the fungal genus level. A variety of *Fusarium* is a vital plant pathogen, and *Fusarium* is the most prominent strain producing DON [41]. This is reflected in the DON content of the feed. The DON content was significantly lower in the Q1-2 and Q27-2 groups compared to the control groups, and it met the legal limit of 1.00 mg/kg in feed products in China. As for AFB1, each treatment group met the Chinese feed product limit of 10 µg/kg, and the AFB1 content of strain Q1-2 was the lowest among all treatment groups. *Aspergillus* is usually closely associated with AFB1 production [42], and this was verified by the positive correlation between the relative abundance of *Aspergillus* and AFB1 content in this study. The sudden increase in the relative abundance of *Aspergillus* in the Q27-2 group at 30 d may be due to the fact that as the pH of the feed increased, it shook the dominance of *Lactiplantibacillus* in the feed, thus promoting the growth of undesirable microorganisms. On the other hand, it is possible that the reduction in fungal growth inhibition was caused by the depletion of inhibitory compounds produced by Q27-2. Interestingly, AFB1 contents were not as significantly elevated as the relative abundance of *Aspergillus* in this treatment group. This may be due to the fact that Q27-2 reduces AFB1 content mainly by inhibiting AFB1 secretion by *Aspergillus*. This did not occur in the group with the addition of the strain Q1-2, and the treatment group was almost free of *Aspergillus* at the end of fermentation, suggesting that the strain Q1-2 not only reduces mycotoxins but also effectively inhibits fungal growth. In the practical applications of feeds, where aerobic exposure is often required, treatment groups with the fungi inhibiting effects of LAB may have better feed quality in actual production [43]. In fact, the inhibition of the growth of associated fungi is not a sufficient condition for a reduction in mycotoxin content. Some studies demonstrated that AFB1 levels increase as *Aspergillus* growth is inhibited—i.e., aflatoxins are the stress-related toxins [44]. However, based on the results presented in the heat map, there was a significant positive correlation between the mycotoxin content and the relative abundance of the fungus. This may be due to the fact that the inhibition of fungal growth to a level where mycotoxin cannot be produced is followed by a decrease in mycotoxin content and a concomitant decrease in its toxicity, which is consistent with the results of the study by Sadeghi et al. [45]. The reasons associated with the reduction in feed mycotoxin levels are relatively complex and often result from the interaction of multiple causes. In addition to the above causes, there is also the possibility that LAB adsorb or degrade mycotoxins, as both strains of *Lactiplantibacillus* have shown the ability to reduce mycotoxin levels in vitro. Hernandez-Mendoza et al. [21] showed that LAB can mitigate mycotoxin contamination in feed by adsorbing AFB1 to reduce the absorption of the toxin in the animal’s intestine. Even though the degradation of AFB1 by LAB may produce substances that are still toxic, the toxicity of the feed is reduced, as demonstrated in the study by Zuo et al. [46]. Although it is a challenge to confirm from a single test what the cause of the reduction in mycotoxins in feed is, the aim of mitigating fungal and mycotoxin contamination and improving feed quality by the addition of LAB has been achieved. This series of results demonstrates that the addition of strains Q1-2 and Q27-2 could improve feed quality and reduce mycotoxins contents, indicating that it could be used as an additive in the production of fermented feeds.

## 4. Conclusions

This study confirmed that the inoculation of the screened *L. plantarum* Q1-2 and *L. salivarius* Q27-2 had positive effects on the fermentation process. Adding LAB could significantly improve the fermentation process by increasing the abundance of *Lactiplantibacillus* and the content of lactic acid, thus reducing the pH value. The Q1-2 and Q27-2 groups exhibited significantly lower contents of mycotoxins than the control group, with a particularly significant reduction in DON contents. Furthermore, adding the strain Q1-2 could reduce the relative abundance of *Fusarium* and *Aspergillus*. In summary, strains Q1-2 and Q27-2 are the appropriate feed additive choice. This study provides a preliminary reference for practical production in the field of feed and food preservation. However, the mechanism of their reduction of fungi and mycotoxins needs to be further studied.

## 5. Materials and Methods

### 5.1. LAB sample Collection and Isolation

Feces were obtained from healthy pigs from the Huzhu County Pig Breeding Plant (Xining, China). Upon collection, the samples were promptly stored in sterile plastic bags, frozen in liquid nitrogen, and transported in dry ice to the Henan Provincial Key Laboratory of Ion Beam Bioengineering. LAB strains were isolated from feces samples by using de Man, Rogosa, and Sharpe (MRS) agar medium at 37 °C. The cream-white raised colonies with different shapes and sizes were chosen for the Gram staining test, and gram-positive isolates were purified and stored at −80 °C for further assays.

### 5.2. Screening of LAB

#### 5.2.1. Screening for Antibacterial LAB Strains

The Oxford cup double-layer plate method outlined by Chen et al. [47] was used to screen LAB strains with broad-spectrum antibacterial activity. A total of 8 strains preserved in our laboratory were selected as pathogenic indicator bacteria, including *Escherichia coli* ATCC 30105; *Micrococcus luteus* ATCC 4698; *Staphylococcus aureus* ATCC 29213; *Pseudomonas aeruginosa* ATCC 27853; *Listeria monocytogenes* BAA; *Bacillus subtilis* ATCC 6633; *Salmonella enterica subsp. enterica serovar* Enteritidis ATCC 13076; and *Salmonella enterica subsp. enterica serovar* Typhimurium ATCC 43971. To evaluate the inhibitory activity, 200 µL of LAB cell-free supernatant was added to each Oxford cup and incubated at 37 °C for 24 h. The diameter of the inhibition zone was measured to compare the efficacy of the strains.

#### 5.2.2. Screening for Antifungal LAB Strains

After activation of the LAB, the antifungal ability of the strains was evaluated using the double plate method of Magnusson et al. [48]. When the bilayer plate was thoroughly solidified, it was transferred to a constant temperature incubator at 37 °C for 24 h. Evaluation of the antifungal effect of LAB strains according to the diameter of the inhibition circle was carried out.

#### 5.2.3. Screening LAB strains for Mycotoxin Removing Capacity

Reduction rate of AFB1 and DON by LAB strains was determined using a slightly modified method provided by Hernandez-Mendoza et al. [21]. The organic solvent that had been dissolved with AFB1 and DON standards was added to the PBS solution. The volume was fixed to a toxin concentration of 5 µg/mL and employed as a working solution for use. The LAB strain was inoculated into the MRS liquid medium with 1% inoculum volume and incubated at 37 °C for 12 h. The concentration of LAB was adjusted to 1 × 10^10^ colony-forming units per milliliter (cfu/mL). The culture was added to sterilized centrifuge tubes on a sterile clean table, centrifuged (10 °C, 4000 r/min, 15 min) in a high-speed refrigerated centrifuge (Beckman Coulter, Brea, CA, USA); the supernatant was poured off and the precipitate was washed three times with phosphate-buffered saline (pH 7.2), then the washed precipitate was added to the AFB1 and DON working solution, then incubated at 37 °C for 24 h, centrifuged under the same conditions as above, and the supernatant was stored at −20 °C. The levels of AFB1 and DON were analyzed using enzyme-linked immunosorbent assay (ELISA) kits supplied by Lianshuo Biological Technology Co., Ltd. (AMEKO, Shanghai, China). The microplate reader was utilized to measure the absorbance at a wavelength of 450 nm, and standard curves were drawn according to the external mycotoxin standards provided by the kit for calculating mycotoxin content. The control group was AFB1 and DON working solution without LAB, and the control and treatment groups had three replicates each.

#### 5.2.4. Adsorption Capacity of Lactic Acid Bacteria Inactivated by Heat

To verify the pathway by which mycotoxins were reduced by LAB, after the strain was inactivated by heating at 121 °C for 30 min, the mycotoxin adsorption rate was determined by the method in Section 5.2.3.

### 5.3. DNA Extraction and 16S rRNA Gene Amplification and Sequencing

The extraction of DNA was performed, and the amplification and sequencing of the 16S rRNA gene were carried out in accordance with the protocol described by Wang et al. [49] with minor modifications. The bacterial DNA extraction kit D3350-02 (Omega Biotek, Norcross, GA, USA) was used to extract the DNA of the LAB, and the extracted DNA was used as the template. Bacterial 16S rRNA universal primers were used for the PCR reaction. PCR reaction system: Premix Taq 12.5 µL, DNA template 0.5 µg, 27 F, and 1492 R primers (27F:5′-AGAGTTTGATCCTGGCTCAG-3′, 1492R:5′-GGTTACCTTGTTACGACTT-3′), 0.25 µM each, and ddH_2_O were added to 25 µL. PCR amplification procedure: initial activation at 95 °C for 3 min; 35 cycles at 95 °C for 15 s, 60 °C for 15 s, and 72 °C for 90 s; and a final cycle at 72 °C for 15 min. The PCR products were analyzed using 1% agarose gel electrophoresis and subsequently submitted to Huada Biotech Company Co., Ltd. (Zhengzhou, China) for sequencing. The splice sequences were analyzed by BLAST on the NCBI website (https://blast.ncbi.nlm.nih.gov (accessed on 3 November 2022)); the Kimura2-parameter method in the MEGA7.0 software (https://www.megasoftware.net (accessed on 7 November 2022)) was used to calculate the genetic distance between target sequences and sequences with high homology, and the phylogenetic tree was constructed using the neighbor-joining method.

### 5.4. Biological Characteristics of LAB Strains

To further evaluate the characteristics of the LAB strains, growth curve measurements, and physiological and biochemical tests were carried out on the strains. Individual LAB colonies were selected and added to 20 mL of sterile MRS medium. The optical density at 600 nm (OD 600) was measured at 2 h intervals (2, 4, 6, 8, 10, 12, 14, 16, 18, 20, 22, and 24 h) following inoculation at 37 °C.

The growth of LAB strains was detected under different pH, temperature, and NaCl conditions. Strains were cultured in liquid MRS medium at 5 °C, 10 °C, 30 °C, 45 °C, and 50 °C for 7 days for the growth temperature test. MRS medium with different pH gradients (including pH 3.0, 3.5, 4.0,5.0,6.0,8.0, 9.0, and 10.0) was cultured at 37 °C for 7 days, and the results were recorded. NaCl tolerance of strains was measured in liquid MRS medium containing 4% (*w*/*v*) and 7% (*w*/*v*) NaCl for 2 days.

### 5.5. Fermentation Feed Production

The alfalfa for this study was purchased from Spreadtrum Biotechnology Co., Ltd. (Xi’an, China). Based on the ratio of Table 6, 1–2 cm alfalfa was mixed with corn, wheat bran, and other concentrates. After the moisture content was adjusted to 55%, 1 × 10^6^ cfu/mL strain Q1-2 and strain Q27-2 were added into the pre-fermented feed. The experimental treatment design was as follows: (1) CK (control check); (2) CK + Q1-2; (3) CK + Q27-2. After the feed was well mixed, it was packed into polyethylene bags of 500 g per bag and sealed confidentially with a vacuum sealing machine (P-290, Shineye, Dongguan, China). The experimental design included triplicate treatments, resulting in a total of 45 bags. Samples were collected from each treatment group in 1, 3, 7, 15, and 30 days to determine microbiological and nutritional changes.

### 5.6. Chemical Composition Analysis

After sampling on 1, 3, 7, 15, and 30 days, the feed was immediately stored in sealed sample bags at −20 °C for further analysis. The 10 g sample was dissolved in 90 mL of deionized water, and the mixture was subjected to vortex mixing for 30 min. The pH value of the resulting solution was measured using a pH meter (Mettler Toledo Co., Ltd., Greifensee, Switzerland). The analysis of its chemical composition has been reported elsewhere by Li et al. [50]. At each opening, a 10 g sample was mixed with 90 mL of distilled water and filtered with 4-layer nylon gauze and qualitative filter paper. The diluted samples were then determined for organic acid concentrations by using high-performance liquid chromatography (HPLC). The ammonia nitrogen content of the fermentation samples was determined using the phenol-sodium hypochlorite colorimetric method. The dry matter (DM) was determined after drying the samples in an oven at 65 °C for 48 h. The Automatic Kjeldahl Apparatus (K9860, Hainon, Shandong, China) was used for determining the crude protein (CP) content. The polyester mesh bag method was used to determine the counts of neutral detergent fiber (NDF) and acid detergent fiber (ADF).

### 5.7. Bacterial and Fungal Community Analysis

The DNA extraction and polymerase chain reaction amplification were performed in accordance with the methodology outlined by Zhou et al. [51]. The total genomic DNA was extracted using the CTAB/SDS method, and the quality of the extracted DNA was evaluated by 1% agarose gel electrophoresis. PCR amplification of the V3–V4 region of 16S rRNA for bacteria and ITS1F region for fungi was performed using the extracted DNA samples. The PCR products were purified using the AMPure XP system after electrophoresis on 2% agarose gels to obtain the original library of the samples. Subsequently, the DNA samples were sequenced using the MiSeq platform (Shanghai Tianhao Biotechnology Co., LTD., China). To ensure high-quality sequences, all paired readings were processed with FLASH2 (version 1.2.11) and filtered according to the QIIME quality control process (version 1.9.1). The R software package (version 2.15.3) was used for beta diversity analysis.

### 5.8. Determination of AFB1 and DON Mycotoxins in Fermented Feed

The dried and crushed sample (5 g) was mixed with 10 mL of 70% methanol. The mixture was shaken vigorously for 3 min and then filtered through a filter paper. The filtrate was collected and used for mycotoxin content determination using ELISA, as described in Section 5.2.3.

### 5.9. Statistical Analysis

In this experiment, SPSS 20.0 software was used for data processing and statistical analysis, Origin 2017 was used for drawing, and one-way ANOVA was used to test and analyze the significance of differences (*p* < 0.05). The data of high-throughput sequencing were analyzed on the online platform https://magic.novogene.com (accessed on 21 December 2022). Each sample was tested for chemical composition three times, and the mean value was reported as the result.

## Figures and Tables

**Figure 1 toxins-15-00226-f001:**
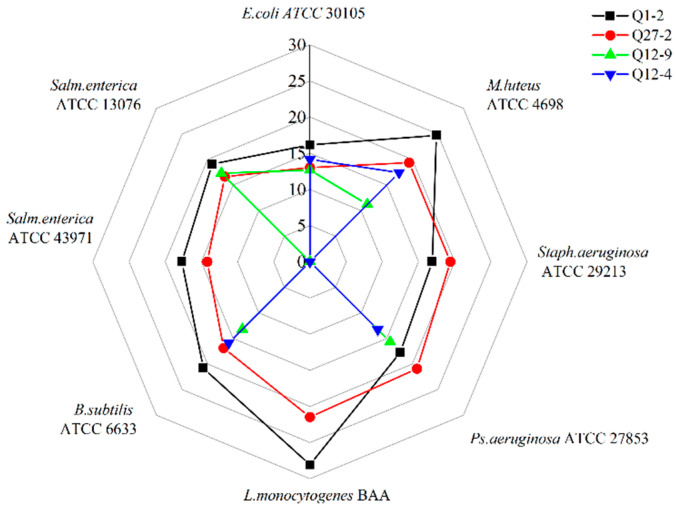
The antibacterial activity of strains. The value represents the diameter of the inhibition zone. The diameter of the Oxford cup was 10.00 mm.

**Figure 2 toxins-15-00226-f002:**
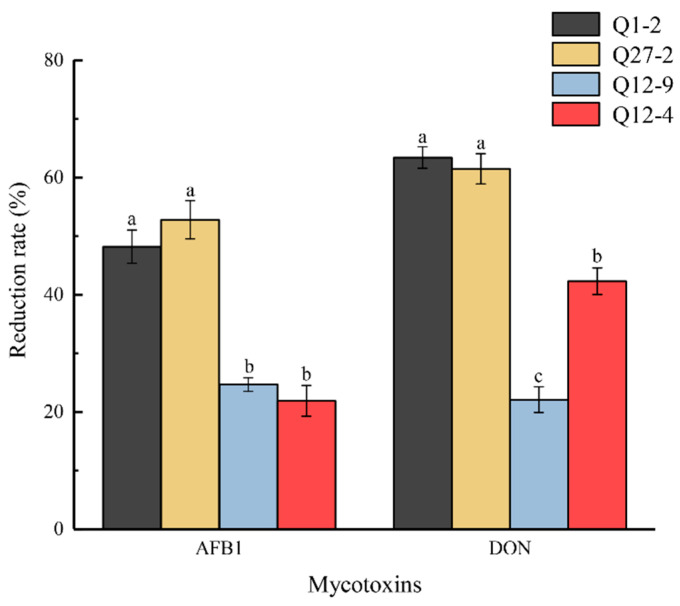
Reduction effect of strains on mycotoxins. Different lowercase letters (a–c) for the same mycotoxin indicate significant differences (*p* < 0.05).

**Figure 3 toxins-15-00226-f003:**
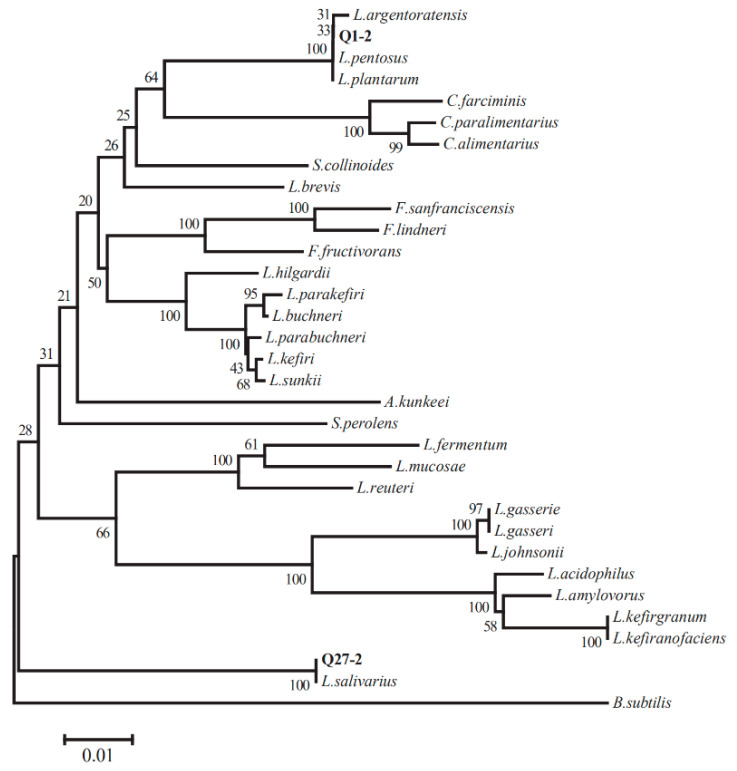
Phylogenetic tree showing the relative position of strains Q1-2 and Q27-2 by the neighbor-joining method with 16S rRNA gene sequences.

**Figure 4 toxins-15-00226-f004:**
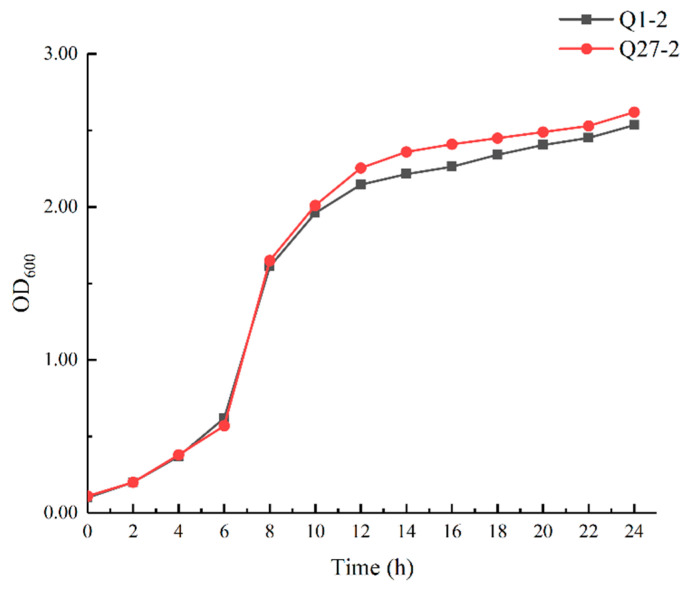
The growth curve of two lactic acid bacteria isolates cultured at 37 °C.

**Figure 5 toxins-15-00226-f005:**
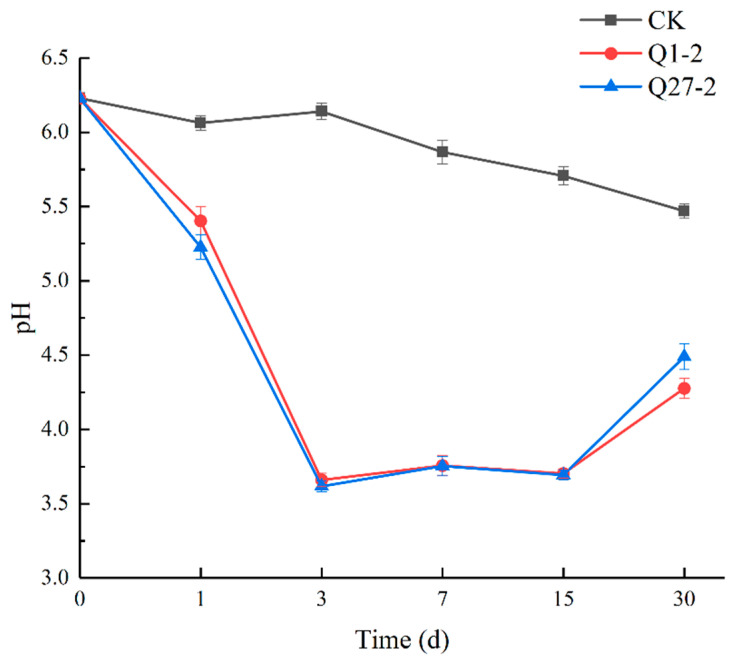
Changes in pH value during fermentation.

**Figure 6 toxins-15-00226-f006:**
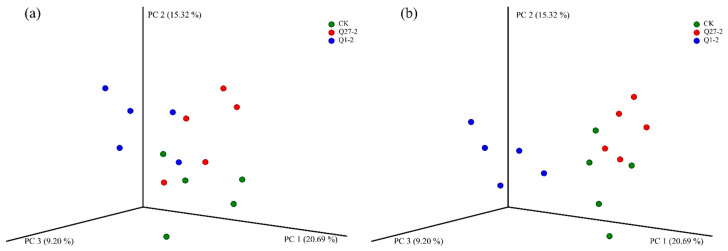
Principal coordinate analysis plots of bacterial (**a**) and fungal (**b**) compositions of different treated feeds during fermentation. Different circles in the same color represent samples of different periods.

**Figure 7 toxins-15-00226-f007:**
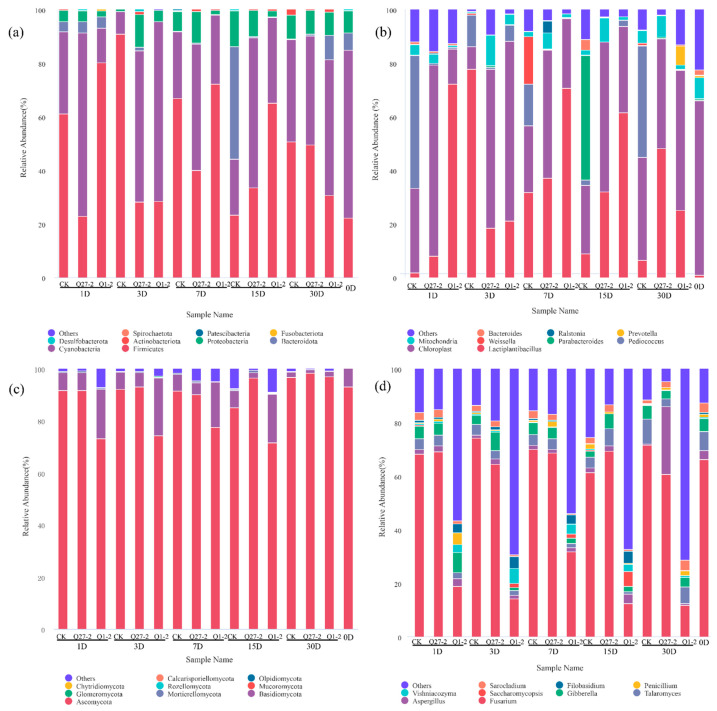
The bacterial and fungal communities of feed during fermentation. The phylum level of bacterial and fungal communities was shown in (**a**,**c**); and the genus level was shown in (**b**,**d**). d, days.

**Figure 8 toxins-15-00226-f008:**
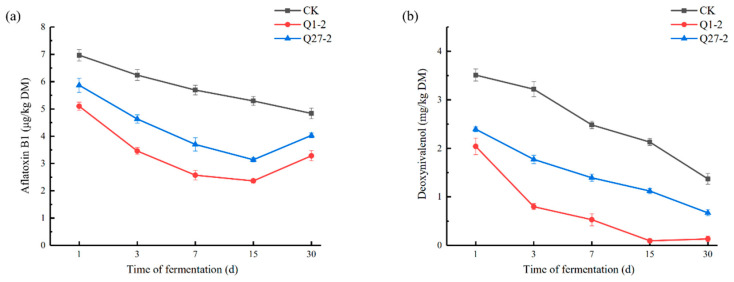
Content of mycotoxins during fermentation. (**a**) aflatoxin B1 (µg/kg DM); (**b**) deoxynivalenol (mg/kg DM).

**Figure 9 toxins-15-00226-f009:**
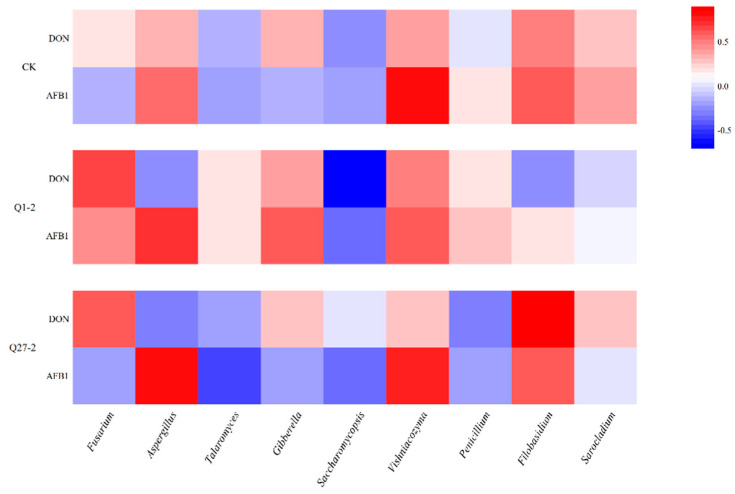
Spearman correlation heatmap of fungi at the genus level with mycotoxins during fermentation.

**Table 1 toxins-15-00226-t001:** The antifungal activity of strains Q1-2, Q27-2, Q12-9, and Q12-4.

Indicators	Q1-2	Q27-2	Q12-9	Q12-4
*Aspergillus niger*	+	++	++	++
*Aspergillus flavus*	++	+	−	−
*Aspergillus oryzae*	+	+	++	−
*Penicillium citrus*	+	++	+	++
*Trichoderma*	++	+	−	+

−: no inhibitory effect; +: an inhibition zone ranging from 0.1% to 5% of the total plate area; ++:an inhibition zone accounting for more than 5% of the total plate area.

**Table 2 toxins-15-00226-t002:** Mycotoxins binding by different LAB strains.

Strains	AFB1 Binding (% of Total)		DON Binding (% of Total)	
	Viable	Heat Inactivation	Viable	Heat Inactivation
Q1-2	48.21 ± 2.83c	56.33 ± 1.60b	63.40 ± 1.83a	64.27 ± 1.22a
Q27-2	52.82 ± 3.24c	53.21 ± 1.57c	61.52 ± 2.59b	69.36 ± 0.95a

Values were expressed as means ± SEM (*n* = 3), and values within the same row with different letters are significantly different (*p* < 0.05).

**Table 3 toxins-15-00226-t003:** The results of physiological and biochemical characteristics of strains.

Strains	Concentration of NaCl (%)		Temperature (°C)					pH							
	4	7	5	10	30	45	50	3	3.5	4	5	6	8	9	10
Q1-2	+	+	+	+	+	w	−	w	+	+	+	+	+	+	+
Q27-2	w	+	+	+	+	+	w	−	w	+	+	+	+	+	+

+: normal growth; w: weak growth; −: no growth.

**Table 4 toxins-15-00226-t004:** Fermentation indexes of different fermentation groups at different periods.

Items	Treatment	Fermenting Days					SEM	*p*-Value		
		1 d	3 d	7 d	15 d	30 d		T	D	T × D
LA (%DM)	CK	0.26 ^Ac^	0.23 ^Bc^	0.64 ^Cc^	1.33 ^Cb^	2.56 ^Ba^	0.149	<0.05	NS	NS
	Q1-2	0.25 ^Ae^	1.29 ^Ad^	1.92 ^Ac^	2.55 ^Ab^	4.06 ^Aa^				
	Q27-2	0.26 ^Ad^	1.17 ^Acd^	1.51 ^Bc^	2.09 ^Bb^	4.26 ^Aa^				
AA (%DM)	CK	ND	0.01	0.31	0.66	1.49 ^A^	0.049	<0.05	<0.05	<0.05
	Q1-2	ND	0.06	0.25	0.37	0.74 ^B^				
	Q27-2	ND	0.07	0.14	0.27	0.38 ^C^				
PA (%DM)	CK	ND	0.31	0.27	0.16	0.55	-	-	-	-
	Q1-2	ND	0.09	ND	ND	ND				
	Q27-2	ND	0.17	ND	ND	ND				
BA (%DM)	CK	0.15	0.24	0.34	0.55	0.71	-	-	-	-
	Q1-2	ND	0.12	0.07	ND	ND				
	Q27-2	ND	0.09	ND	ND	ND				
NH_3_–N (g/kg DM)	CK	5.81 ^Ab^	6.55 ^Aa^	5.16 ^Ac^	5.34 ^Ac^	5.15 ^Ac^	0.081	<0.05	<0.05	NS
	Q1-2	4.75 ^Bb^	5.39 ^Ba^	4.74 ^Bb^	4.27 ^Bc^	3.83 ^Bc^				
	Q27-2	4.37 ^Cb^	4.64 ^Ca^	3.84 ^Cc^	3.55 ^Cd^	3.31 ^Be^				

LA, lactic acid; AA, acetic acid; PA, propionic acid; BA, butyric acid; NH_3_–N, ammonia nitrogen; ND, not detected; SEM, standard error of the mean; T, treatments; D, fermenting days; T × D, the interaction between treatments and fermenting days. The means in the same column (A–C) or row (a–e) with different superscript letters differ significantly from each other (*p* < 0.05).

**Table 5 toxins-15-00226-t005:** Chemical indexes of fermentation feed at different stages in each treatment group.

Items	Treatment	Fermenting Days					SEM	*p*-Value		
		1 d	3 d	7 d	15 d	30 d		T	D	T × D
DM (%FM)	CK	42.58	42.46	43.07	43.10	42.80	0.7110	NS	NS	NS
	Q1-2	43.98	43.01	43.51	42.58	43.92				
	Q27-2	42.28	42.95	43.11	42.04	42.09				
CP (%DM)	CK	19.47	19.03	18.82	18.27	17.92	0.0670	<0.05	<0.05	<0.05
	Q1-2	19.27	18.80	18.36	17.71	17.19				
	Q27-2	19.31	18.78	18.21	17.39	16.86				
NDF (%DM)	CK	50.71	45.12	44.30	44.00	41.63	0.6400	NS	<0.05	NS
	Q1-2	50.65	46.68	44.62	44.60	41.88				
	Q27-2	50.52	48.80	44.55	44.19	42.07				
ADF (%DM)	CK	35.69	31.50	28.82	27.58	26.76	0.3180	NS	<0.05	NS
	Q1-2	36.23	31.06	29.03	27.50	26.47				
	Q27-2	36.16	31.90	28.99	27.55	26.63				

FM, fresh material; DM, dry matter; CP, crude protein; NDF, neutral detergent fiber; ADF, acid detergent fiber; WSC, water-soluble carbohydrates; NS, not significant; SEM, standard error of the mean; T, treatments; D, fermenting days; T × D, the interaction between treatments and fermenting days.

**Table 6 toxins-15-00226-t006:** Ingredients and proportions of various components in the pre-fermentation feed.

Items	Ingredients Composition, % of DM
Alfalfa	46.5
Corn	30
Wheat bran	10
Soybean meal	10
Soybean oil	2
Premixture	1
Sodium chloride	0.5
Total	100

## Data Availability

The 16Sr RNA gene sequences of strains Q1-2, and Q27-2 used to support the findings of this study have been deposited in the GenBank repository with accession numbers OQ422573 and OQ422574, respectively.

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
