# Peer review of "Effects of Lactic Acid Bacteria Reducing the Content of Harmful Fungi and Mycotoxins on the Quality of Mixed Fermented Feed"

_toxins, 2023, doi:10.3390/toxins15030226_

Round 1

Reviewer 1 Report

The paper “Effects of lactic acid bacteria reducing the content of harmful fungi and mycotoxins on the quality of mixed fermented feed” reports the identification of lactic acid bacteria with positive effects on the fermentation process and capable to reduce the gro. In this work the authors identified two strains, Lactobacillus salivarius Q27-2 and Lactobacillus plantarum Q2-1, with strong antibacterial and fungicidal activity, and able to reduce B1 aflatoxin and DON content. In addition, the authors stated that these two strains have a positive effect on the fermentation index.

In my opinion, this manuscript is well written, with a large quantity of well-presented data. The work is interesting and appropriate to be published in Toxins Journal.

Some suggestions to improve the manuscript

Lines 23-34: In the sentence the word “contamination” is repeated three times. Please, improve this sentence.

Line 48: Please, add a space between ”[9]” and “and”

Line 50: Please, write the full name of L. rhamnosus

Line 71: Please, write the full name of Pediococcus

Line 131: Please, write the sequence of primer F with all uppercase characters

Line 132: how much DNA was used? 1 µl of what? Please, specify the DNA quantity in ng

Line 133: how much primer was used? 1 µl of what? Please, specify the primer concentration

Line 142: Strain of what? Bacterial? Please, complete the paragraph title

Line 189: Please, write MiSeq

Line 193: Please, delete the word “in” between DON and Mycotoxins

Lines 194-196: I suggest improving this sentence. For example: 5 g of dried and crushed sample were mixed with 10 mL of 70% methanol, the mixture was shaken vigorously for 3 min, and filtered through a filter paper (Please, specify the type of filter), and the filtrate was collected.

Line 196: “The filtrate was collected and filtered through filter paper” is it a repetition?

Figure 1: In the legend of the figure there is an error, I think the red indicates the strain Q27-2 and not Q27-1. Please, correct

Line 222: Please, correct the paragraph title

Line 248: paragraph 3.6? Where is paragraph 3.5? Please, correct

Line 248: Table 5? Where is table 4? Please, correct

Figure 6: it is difficult to read the writing on the axes. Please, improve the resolution

Figure 7: it is difficult to read. Please, improve the resolution

Pages 13-14: References are not in order. Reference [33] comes after [36]. Please, correct

Line 418: Please, write CO2 with 2 in subscript form

Line 420: Please, add a space after ”[38]”

Lines 486-596: Please, write the name of Journal in abbreviated form (as reported inInstructions for Authors”: https://www.mdpi.com/journal/toxins/instructions)

Reviewer 2 Report

The topic and results could be interesting for scientific community, but in the actual form the paper can not be accepted.

The introduction should be improved. It should be clear to the reader why the choice of the two mycotoxins, AF and DON, why not some other combination (I presume because AF B1 is the most toxic and DON is among the most frequent)? Also it should be explained, briefly, what are mycotoxins. I should mention by the toxic effects that AF B1 is a mutagen and carcinogen. Please reformulate the text playing attention that the meaning is clear. Mycontoxin are not detected only in food and feed, but also in raw materials; antibacterial activity of some compound does not mean that it is surely also antifungal.  Ergot is not a fungal genera, it is a disease (pathology) caused by fungi from Claviceps genera. Please decide which taxonomy you are using for your LAB and use always the same one (or Lactiplantibacillus plantarum or Lactobacillus plantarum, not once one then the other. 

In Material and methods (subchapter 2.2.2) correct antibacterial in antifungal (line 105) 

The results are correctly reported. I just wonder why you haven't verify the adsorption of my toxins. In the introduction you talk about degradation and adsorption, but in the research you don't verify adsorption.

In the discussion you should keep in mind that impaired growth of fungus dams not mean always toxin inhibition, AF for example are stress related toxins and different papers report that impaired growth resulted in higher AF production. Furthermore, the rise of AF after 30 days could also be related to the consumption of inhibitors. It is known that some antioxidants inhibit AF synthesis (and some also A. flavus growth) but after they are consumed there is a rise of AF production. Another thing that you should comment is that you haven't discussed is a potential toxicity of the product of mycotoxin degradation. It is true that generally products of degradation are less toxic than the original toxin, but can still be toxic (i.e. AF M1).

Round 2

Reviewer 2 Report

The revision improved the quality of the paper. There is just one small thing, in material and methods you have a subchapter 5.2.3. Screening for LAB strains with mycotoxin degrading ability, and the description suites to removal of mycotoxins by LAB (could be adsorption but also degradation, it is cleared by a successive experiment). I suggest to rename it in "Screening  LAB strains for mycotoxin removing capacity" or similar. 

In my opinion the paper can be published